# GENERATIVE MODELS FOR GRAPH-BASED PROTEIN DESIGN

**John Ingraham, Vikas K. Garg, Regina Barzilay, Tommi Jaakkola**
CSAIL, MIT

## ABSTRACT

Engineered proteins offer the potential to solve many problems in biomedicine, energy, and materials science, but creating designs that succeed is difficult in practice. A significant aspect of this challenge is the complex coupling between protein sequence and 3D structure, and the task of finding a viable design is often referred to as the *inverse protein folding problem*. We develop generative models for protein sequences conditioned on a graph-structured specification of the design target. Our approach efficiently captures the complex dependencies in proteins by focusing on those that are long-range in sequence but local in 3D space. Our framework significantly improves upon prior parametric models of protein sequences given structure, and takes a step toward rapid and targeted biomolecular design with the aid of deep generative models.

## 1 INTRODUCTION

A central goal for computational protein design is to automate the invention of protein molecules with defined structural and functional properties. This field has seen tremendous progess in the past two decades (Huang et al., 2016), including the design of novel 3D folds (Kuhlman et al., 2003), enzymes (Siegel et al., 2010), and complexes (Bale et al., 2016). However, the current practice often requires multiple rounds of trial-and-error, with first designs frequently failing (Koga et al., 2012; Rocklin et al., 2017). Several of the challenges stem from the bottom-up nature of contemporary approaches that rely on both the accuracy of *energy functions* to describe protein physics as well as on the efficiency of *sampling algorithms* to explore the protein sequence and structure space.

Here, we explore an alternative, top-down framework for protein design that directly learns a conditional generative model for protein *sequences* given a specification of the target structure, which is represented as a *graph* over the sequence elements. Specifically, we augment the autoregressive self-attention of recent sequence models (Vaswani et al., 2017) with graph-based descriptions of the 3D structure. By composing multiple layers of structured self-attention, our model can effectively capture higher-order, interaction-based dependencies between sequence and structure, in contrast to previous parameteric approaches (O'Connell et al., 2018; Wang et al., 2018) that are limited to only the first-order effects.

The graph-structured conditioning of a sequence model affords several benefits, including favorable computational efficiency, inductive bias, and representational flexibility. We accomplish the first two by leveraging a well-evidenced finding in protein science, namely that long-range dependencies in sequence are generally short-range in 3D space (Marks et al., 2011; Morcos et al., 2011; Balakrishnan et al., 2011). By making the graph and self-attention similarly sparse and localized in 3D space, we achieve computational scaling that is linear in sequence length. Additionally, graph structured inputs offer representational flexibility, as they accomodate both coarse, 'flexible backbone' (connectivity and topology) as well as fine-grained (precise atom locations) descriptions of structure.

We demonstrate the merits of our approach via a detailed empirical study. Specifically, we evaluate our model at *structural generalization* to sequences of protein folds that were outside of the training set. Our model achieves considerably improved generalization performance over the recent deep models of protein sequence given structure as well as structure-naïve language models.

## 1.1 RELATED WORK

**Generative models for proteins**  A number of works have explored the use of generative models for protein engineering and design (Yang et al., 2018). Recently O'Connell et al. (2018) and Wang et al. (2018) proposed neural models for sequences given 3D structure, where the amino acids at different positions in the sequence are predicted independently of one another. Greener et al. (2018) introduced a generative model for protein sequences conditioned on a 1D, context-free grammar based specification of the fold topology. Boomsma & Frellsen (2017) and Weiler et al. (2018) used deep neural networks to model the conditional distribution of letters in a specific position given the structure and sequence of all surrounding residues. In contrast to these works, our model captures the joint distribution of the full protein sequence while grounding these dependencies in terms of long-range interactions arising from the structure.

In parallel to the development of structure-based models, there has been considerable work on deep generative models for protein sequences in individual protein families with directed (Riesselman et al., 2018; Sinai et al., 2017) and undirected (Tubiana et al., 2018) latent variable models. These methods have proven useful for protein engineering, but presume the availability of a large number of sequences from a particular family.

More recently, several groups have obtained promising results using unconditional protein language models (Bepler & Berger, 2019; Alley et al., 2019; Heinzinger et al., 2019; Rives et al., 2019) to learn protein sequence representations that can transfer well to supervised tasks. While serving different purposes, we emphasize that one advantage of conditional generative modeling is to facilitate adaptation to specific (and potentially novel) parts of structure space. Language models trained on hundreds of millions of evolutionary sequences are unfortunately still 'semantically' bottlenecked by the much smaller number of evolutionary 3D folds (perhaps thousands) that the sequences design. We propose evaluating protein language models with *structure*-based splitting of sequence data (Section 3, albeit on much smaller sequence data), and begin to see how unconditional language models may struggle to assign high likelihoods to sequences from out-of-training folds.

In a complementary line of research, deep models of protein structure (Anand & Huang, 2018; Ingraham et al., 2019; AlQuraishi, 2018) have been proposed recently that could be used to craft 3D structures for input to sequence design.

**Protein design**  For classical approaches to computational protein design, which are based on joint modeling of structure and sequence, we refer the reader to a review of both methods and accomplishments in Huang et al. (2016). More recently, Zhou et al. (2018) proposed a non-parametric approach to protein design in which a target design is decomposed into substructural motifs that are then queried against a protein database. In this work we will focus on comparisons with direct parametric models of the sequence-structure relationship.

**Self-Attention**  Our model extends the Transformer (Biswas et al., 2018) to additionally capture sparse, pairwise relational information between sequence elements. The dense variation of this problem was explored in Shaw et al. (2018) and Huang et al. (2018). As noted in those works, incorporating general pairwise information incurs $\mathcal{O}(N^2)$ memory (and computational) cost for sequences of length $N$, which can be highly limiting for training on GPUs. We circumvent this cost by instead restricting the self-attention to the sparsity of the input graph. Given this graph-structured self-attention, our model may also be reasonably cast in the framework of message-passing or graph neural networks (Gilmer et al., 2017; Battaglia et al., 2018). Our approach is similar to Graph Attention Networks (Veličković et al., 2017), but augmented with edge features and an autoregressive decoder.

## 2 METHODS

### 2.1 REPRESENTING STRUCTURE

We represent protein structure in terms of an attributed graph $\mathcal{G} = (\mathcal{V}, \mathcal{E})$ with node features $\mathcal{V} = \{\boldsymbol{v}_1, \ldots, \boldsymbol{v}_N\}$ and edge features $\mathcal{E} = \{\boldsymbol{e}_{ij}\}_{i \neq j}$ over the sequence residues (amino acids). This formulation can accommodate different variations on the macromolecular design problem, including

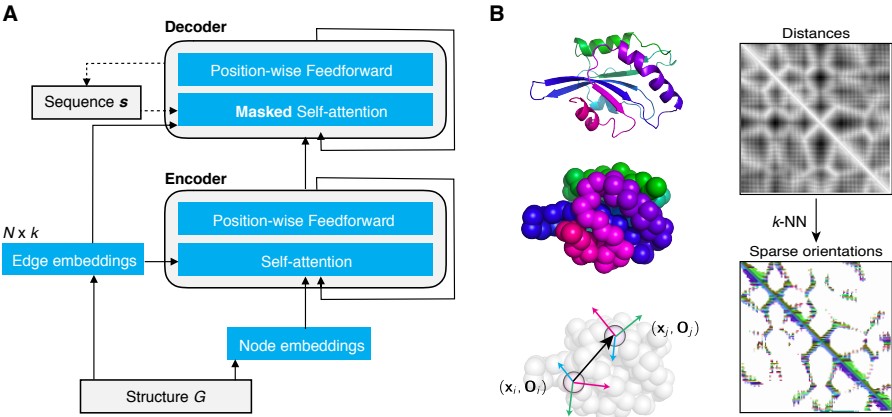

Figure 1: **An autoregressive self-attention model for protein sequences given 3D structures.**
(A) The encoder develops position-wise representations of structure using multi-head self-attention
(Vaswani et al., 2017) over nodes and edges of an input graph. The attention heads are structured by
the sparsity of the input graph, enabling efficient computation for large molecules with thousands
of atoms (See Figure 2 for examples). (B) For rigid-body protein design, the graph encodings of
atomic structure (left, top) are based on an encoding of relative positioning of $C_\alpha$ coordinates $\boldsymbol{x}_i$
(left, middle), which are endowed with local coordinate systems $\boldsymbol{O}_i$ based on backbone geometry.
The graph edge features (left, bottom) encode the 6DoF transformations between local coordinate
systems $(\boldsymbol{x}_i, \boldsymbol{O}_i)$ and $(\boldsymbol{x}_j, \boldsymbol{O}_j)$. For sparsity, we derive a $k$-Nearest Neighbors graph from the Eu-
clidean distances (right, top) and restrict all subsequent computation such as orientation calculations
(right, bottom) to this graph.

both the 'rigid backbone' design where the precise coordinates of backbone atoms are fixed, as well
as the 'flexible backbone' design where looser constraints such as blueprints of hydrogen-bonding
connectivity (Koga et al., 2012) or 1D architectures (Greener et al., 2018) could define the structure
of interest.

**3D considerations**    For a rigid-body design problem, the structure for conditioning is a fixed set
of backbone coordinates $\mathcal{X} = \{\boldsymbol{x}_i \in \mathbb{R}^3 : 1 \leq i \leq N\}$, where $N$ is the number of positions[1]. We
desire a graph representation of the coordinates $\mathcal{G}(\mathcal{X})$ that has two properties:

- *Invariance*. The features are invariant to rotations and translations.
- *Locally informative.*    The edge features incident to $\boldsymbol{v}_i$ due to its neighbors $\mathrm{N}(i)$,
  i.e. $\{\boldsymbol{e}_{ij}\}_{j \in \mathrm{N}(i)}$, contain sufficient information to reconstruct all adjacent coordinates
  $\{\boldsymbol{x}_j\}_{j \in \mathrm{N}(i)}$ up to rigid-body motion.

While invariance is motivated by standard symmetry considerations, the second property is mo-
tivated by limitations of current graph neural networks (Gilmer et al., 2017). In these networks,
updates to node features $\boldsymbol{v}_i$ depend only on the edge and node features adjacent to $\boldsymbol{v}_i$. However, typ-
ically, these features are insufficient to reconstruct the relative neighborhood positions $\{\boldsymbol{x}_j\}_{j \in \mathrm{N}(i)}$,
so individual updates cannot fully depend on the 'local environment'. For example, pairwise dis-
tances $D_{ij}$ and $D_{il}$ are insufficient to determine if $\boldsymbol{x}_j$ and $\boldsymbol{x}_l$ are on the same or opposite sides of
$\boldsymbol{x}_i$.

**Structural encodings**    We develop invariant and locally informative features by first augmenting
the points $\boldsymbol{x}_i$ with 'orientations' $\boldsymbol{O}_i$ that define a local coordinate system at each point. We define

---

[1]Here we consider a single representative coordinate per position when deriving edge features but may
revisit multiple atom types per position for features such as backbone angles or hydrogen bonds.

these in terms of the backbone geometry as

$$\boldsymbol{O}_i = [\boldsymbol{b}_i \ \ \boldsymbol{n}_i \ \ \boldsymbol{b}_i \times \boldsymbol{n}_i],$$

where $\boldsymbol{b}_i$ is the negative bisector of angle between the rays $(\boldsymbol{x}_{i-1} - \boldsymbol{x}_i)$ and $(\boldsymbol{x}_{i+1} - \boldsymbol{x}_i)$, and $\boldsymbol{n}_i$ is a unit vector normal to that plane. Formally, we have

$$\boldsymbol{u}_i = \frac{\boldsymbol{x}_i - \boldsymbol{x}_{i-1}}{||\boldsymbol{x}_i - \boldsymbol{x}_{i-1}||}, \quad \boldsymbol{b}_i = \frac{\boldsymbol{u}_i - \boldsymbol{u}_{i+1}}{||\boldsymbol{u}_i - \boldsymbol{u}_{i+1}||}, \quad \boldsymbol{n}_i = \frac{\boldsymbol{u}_i \times \boldsymbol{u}_{i+1}}{||\boldsymbol{u}_i \times \boldsymbol{u}_{i+1}||}.$$

Finally, we derive the spatial edge features $\boldsymbol{e}_{ij}^{(s)}$ from the rigid body transformation that relates reference frame $(\boldsymbol{x}_i, \boldsymbol{O}_i)$ to reference frame $(\boldsymbol{x}_j, \boldsymbol{O}_j)$. While this transformation has 6 degrees of freedom, we decompose it into features for distance, direction, and orientation as

$$\boldsymbol{e}_{ij}^{(s)} = \text{Concat} \left( \mathbf{r} \left( ||\boldsymbol{x}_j - \boldsymbol{x}_i|| \right), \quad \boldsymbol{O}_i^T \frac{\boldsymbol{x}_j - \boldsymbol{x}_i}{||\boldsymbol{x}_j - \boldsymbol{x}_i||}, \quad \mathbf{q} \left( \boldsymbol{O}_i^T \boldsymbol{O}_j \right) \right).$$

Here $\mathbf{r}(\cdot)$ is a function that lifts the distances into a radial basis[2], the term in the middle corresponds to the relative direction of $\boldsymbol{x}_j$ in the reference frame of $(\boldsymbol{x}_i, \boldsymbol{O}_i)$, and $\mathbf{q}(\cdot)$ converts the $3 \times 3$ relative rotation matrix to a quaternion representation. Quaternions represent rotations as four-element vectors that can be efficiently and reasonably compared by inner products Huynh (2009).[3]

**Positional encodings** Taking a cue from the original Transformer model, we obtain positional embeddings $\boldsymbol{e}_{ij}^{(p)}$ that encode the role of local structure around node $i$. Specifically, we need to model the positioning of each neighbor $j$ relative to the node under consideration $i$. Therefore, we obtain the position embedding as a sinusoidal function of the gap $i - j$. Note that this is in contrast to the absolute positional encodings of the original Transformer, and instead matches the relative encodings in Shaw et al. (2018).

**Node and edge features** Finally, we obtain an aggregate edge encoding vector $\boldsymbol{e}_{ij}$ by concatenating the structural encodings $\boldsymbol{e}_{ij}^{(s)}$ and the positional encodings $\boldsymbol{e}_{ij}^{(p)}$ and then linearly transforming them to have the same dimension as the model. We only include edges in the $k$-nearest neighbors graph of $\mathcal{X}$, with $k = 30$ for all experiments.

For node features, we compute the three dihedral angles of the protein backbone $(\phi_i, \psi_i, \omega_i)$ and embed these on the 3-torus as $\{\sin, \cos\} \times (\phi_i, \psi_i, \omega_i)$.

**Flexible backbone features** We also consider 'flexible backbone' descriptions of 3D structure based solely on topological binary edge features. We combine the relative positional encodings with two binary edge features: *contacts* that indicate when the distance between $C_\alpha$ residues at $i$ and $j$ are less than 8 Angstroms and *hydrogen bonds* which are directed and defined by the electrostatic model of DSSP (Kabsch & Sander, 1983). These features implicitly integrate over different 3D backbone configurations that are compatible with the specified topology.

## 2.2 STRUCTURED TRANSFORMER

In this work, we introduce a *Structured Transformer* model that draws inspiration from the self-attention based *Transformer* model (Vaswani et al., 2017) and is augmented for scalable incorporation of relational information. While general relational attention incurs quadratic memory and computation costs, we avert these by restricting the attention for each node $i$ to the set N$(i, k)$ of its $k$-nearest neighbors in 3D space. Since our architecture is multilayered, iterated local attention can derive progressively more global estimates of context for each node $i$. Second, unlike the standard Transformer, we also include edge features to embed the spatial and positional dependencies in deriving the attention. Thus, our model generalizes Transformer to spatially structured settings.

---

[2]We used 16 Gaussian RBFs isotropically spaced from 0 to 20 Angstroms.
[3]We represent quaternions in terms of their vector of real coefficients.

**Autoregressive decomposition** We decompose the joint distribution of the sequence given structure $p(\boldsymbol{s}|\boldsymbol{x})$ autoregressively as

$$p(\boldsymbol{s}|\boldsymbol{x}) = \prod_i p(s_i|\boldsymbol{x}, \boldsymbol{s}_{<i}),$$

where the conditional probability $p(s_i|\boldsymbol{x}, \boldsymbol{s}_{<i})$ of amino acid $s_i$ at position $i$ is conditioned on both the input structure $\boldsymbol{x}$ and the preceding amino acids $\boldsymbol{s}_{<i} = \{s_1, \ldots s_{i-1}\}$ [4]. These conditionals are parameterized in terms of two sub-networks: an *encoder* that computes refined node embeddings from structure-based node features $\mathcal{V}(\mathbf{x})$ and edge features $\mathcal{E}(\mathbf{x})$ and a *decoder* that autoregressively predicts letter $s_i$ given the preceding sequence and structural embeddings from the encoder.

**Encoder** Our encoder module is designed as follows. A transformation $\boldsymbol{W}_h : \mathbb{R}^{d_v} \mapsto \mathbb{R}^d$ produces initial embeddings $\boldsymbol{h}_i = \boldsymbol{W}_h(\boldsymbol{v}_i)$ from the node features $\boldsymbol{v}_i$ pertaining to position $i \in [N] \triangleq \{1, 2, \ldots, N\}$.

Each layer of the encoder implements a multi-head self-attention component, where head $\ell \in [L]$ can attend to a separate subspace of the embeddings via learned query, key and value transformations (Vaswani et al., 2017). The queries are derived from the current embedding at node $i$ while the keys and values from the relational information $\boldsymbol{r}_{ij} = (\boldsymbol{h}_j, \boldsymbol{e}_{ij})$ at adjacent nodes $j \in N(i, k)$. Specifically, $\boldsymbol{W}_q^{(\ell)}$ maps $\boldsymbol{h}_i$ to *query* embeddings $\boldsymbol{q}_i^{(\ell)}$, $\boldsymbol{W}_z^{(\ell)}$ maps pairs $\boldsymbol{r}_{ij}$ to *key* embeddings $\boldsymbol{z}_{ij}^{(\ell)}$ for $j \in \mathrm{N}(i, k)$, and $\boldsymbol{W}_v^{(\ell)}$ maps the same pairs $\boldsymbol{r}_{ij}$ to *value* embeddings $\boldsymbol{v}_{ij}^{(\ell)}$ for each $i \in [N], \ell \in [L]$. Decoupling the mappings for keys and values allows each to depend on different subspaces of the representation.

We compute the attention $a_{ij}^{(\ell)}$ between query $\boldsymbol{q}_i^{(\ell)}$ and key $\boldsymbol{z}_{ij}^{(\ell)}$ as a function of their scaled inner product:

$$a_{ij}^{(\ell)} = \frac{\exp(m_{ij}^{(\ell)})}{\sum\limits_{j' \in \mathrm{N}(i,k)} \exp(m_{ij'}^{(\ell)})}, \qquad \text{where} \quad m_{ij}^{(\ell)} = \frac{\boldsymbol{q}_i^{(\ell)\top} \boldsymbol{z}_{ij}^{(\ell)}}{\sqrt{d}}.$$

The results of each attention head $l$ are collected as the weighted sum $\boldsymbol{h}_i^{(\ell)} = \sum\limits_{j \in N(i,k)} a_{ij}^{(\ell)} \boldsymbol{v}_{ij}^{(\ell)}$ and then concatenated and transformed to give the update $\Delta \boldsymbol{h}_i = \boldsymbol{W}_o \, \mathrm{Concat}\left(\boldsymbol{h}_i^{(1)}, \ldots, \boldsymbol{h}_i^{(L)}\right)$.

We update the embeddings with this residual and alternate between these self-attention layers and position-wise feedforward layers as in the original Transformer (Vaswani et al., 2017). We stack multiple layers atop each other, and thereby obtain continually refined embeddings as we traverse the layers bottom up. The encoder yields the embeddings produced by the topmost layer as its output.

**Decoder** Our decoder module has the same structure as the encoder but with augmented relational information $\boldsymbol{r}_{ij}$ that allows access to the preceding sequence elements $\boldsymbol{s}_{<i}$ in a *causally consistent* manner. Whereas the keys and values of the encoder are based on the relational information $\boldsymbol{r}_{ij} = (\boldsymbol{h}_j, \boldsymbol{e}_{ij})$, the decoder can additionally access sequence elements $s_j$ as

$$\boldsymbol{r}_{ij}^{(\text{dec})} = \begin{cases} (\boldsymbol{h}_j^{(\text{dec})}, \boldsymbol{e}_{ij}, \mathbf{g}(s_j)) & i > j \\ (\boldsymbol{h}_j^{(\text{enc})}, \boldsymbol{e}_{ij}, \mathbf{0}) & i \le j \end{cases}.$$

Here $\boldsymbol{h}_j^{(\text{dec})}$ is the embedding of node $j$ in the current layer of the decoder, $\boldsymbol{h}_j^{(\text{enc})}$ is the embedding of node $j$ in the final layer of the encoder, and $\mathbf{g}(s_j)$ is a sequence embedding of amino acid $s_j$ at node $j$. This concatenation and masking structure ensures that sequence information only flows to position $i$ from positions $j < i$, but still allows position $i$ to attend to subsequent structural information.

---

[4]We anticipate that alternative orderings for decoding the sequence may be favorable but leave this to future work

Table 1: Null perplexities

| Null model | Perplexity | Conditioned on |
|---|---|---|
| Uniform | 20.00 | - |
| Natural frequencies | 17.83 | Random position in a natural protein |
| Pfam HMM profiles | 11.64 | Specific position in a specific protein family |

Table 2: Per-residue perplexities for test sets (lower is better). The test protein structures are cluster-split by CATH topology assignments such that there is no topology (fold) overlap between test, train, and validation.

| Test set | Short | Single chain | All |
|---|---|---|---|
| **Structure-conditioned models** | | | |
| Structured Transformer (ours) | **8.67** | **9.15** | **6.56** |
| SPIN2 (O'Connell et al., 2018) | 12.11 | 12.86 | - |
| **Language models** | | | |
| Structured Transformer, no encoder | 16.03 | 16.36 | 16.98 |
| RNN ($h = 128$) | 16.08 | 16.34 | 16.93 |
| RNN ($h = 256$) | 16.09 | 16.32 | 16.93 |
| RNN ($h = 512$) | 16.01 | 16.34 | 16.94 |
| Test set size | 94 | 107 | 1911 |

We stack three layers of self-attention and position-wise feedforward modules for the encoder and decoder with a hidden dimension of 128 throughout the experiments[5].

## 2.3 TRAINING

**Dataset** To evaluate the ability of the models to generalize across different protein folds, we collected a dataset based on the CATH hierarchical classification of protein structure (Orengo et al., 1997). For all domains in the CATH 4.2 $40\%$ non-redundant set of proteins, we obtained full chains up to length 500 (which may contain more than one domain) and then cluster-split these at the CATH topology level (i.e. fold level) into training, validation, and test sets at an 80/10/10 split. Chains containing multiple CATH tpologies were purged with precedence for test over validation over train. Our splitting procedure ensured that no two domains from different sets would share the same topologies (folds). The final splits contained 18025 chains in the training set, 1637 chains in the validation set, and 1911 chains in the test set.

**Optimization** We trained models using the learning rate schedule and initialization of (Vaswani et al., 2017), a dropout (Srivastava et al., 2014) rate of $10\%$, and early stopping based on validation perplexity.

---

[5]except for the decoder-only language model experiment which used a hidden dimension of 256

Table 3: Test perplexity for different graph features (lower is better).

| Node features | Edge features | Short | Single chain | All |
|---|---|---|---|---|
| **Rigid backbone** | | | | |
| Dihedrals | Distances, Orientations | **8.67** | **9.15** | **6.56** |
| Dihedrals | Distances | 9.33 | 9.93 | 7.75 |
| **Flexible backbone** | | | | |
| - | Contacts, Hydrogen bonds | 11.77 | 12.12 | 11.13 |

## 3 RESULTS

Many protein sequences may reasonably design the same 3D structure (Li et al., 1996), and so we focus on likelihood-based evaluations of model performance. Specifically, we evaluate the perplexity per letter of test protein folds (topologies) that were held out from the training and validation sets.

**Protein perplexities**  What kind of perplexities might be useful? To provide context, we first present perplexities for some simple models of protein sequences in Table 1. The amino acid alphabet and its natural frequencies upper-bound perplexity at 20 and ∼17.8, respectively. Random protein sequences under these null models are unlikely to be functional without further selection (Keefe & Szostak, 2001). First order profiles of protein sequences such as those from the Pfam database (El-Gebali et al., 2018), however, are widely used for protein engineering. We found the average perplexity per letter of profiles in Pfam 32 (ignoring alignment uncertainty) to be ∼11.6. This suggests that even models with high perplexities of this order have the potential to be useful models for the space of functional protein sequences.

**The importance of structure**  We found that there was a significant gap between unconditional language models of protein sequences and models conditioned on structure. Remarkably, for a range of structure-independent language models, the typical test perplexities turned out to be ∼16-17 (Table 2), which were barely better than null letter frequencies (Table 1). We emphasize that the RNNs were not broken and could still learn the training set in these capacity ranges. It would seem that protein language models trained on one subset of 3D folds (in our cluster-splitting procedure) generalize poorly to predict the sequences of unseen folds, which is important to consider when training protein language models for protein engineering and design.

All structure-based models had (unsurprisingly) considerably lower perplexities. In particular, our Structured Transformer model attained a perplexity of ∼7 on the full test set. When we compared different graph features of protein structure (Table 3), we indeed found that using local orientation information was important.

**Improvement over profile-based methods**  We also compared to a recent method SPIN2 that predicts, using deep neural networks, protein sequence profiles given protein structures (O'Connell et al., 2018). Since SPIN2 is computationally intensive (minutes per protein for small proteins) and was trained on complete proteins rather than chains, we evaluated it on two subsets of the full test set: a 'Small' subset of the test set containing chains up to length 100 and a 'Single chain' subset containing only those models where the single chain accounted for the entire protein record in the Protein Data Bank. Both subsets discarded any chains with structural gaps. We found that our Structured Transformer model considerably improved upon the perplexities of SPIN2 (Table 2).

## 4 CONCLUSION

We presented a new deep generative model to 'design' protein sequences given a graph specification of their structure. Our model augments the traditional sequence-level self-attention of Transformers (Vaswani et al., 2017) with relational 3D structural encodings and is able to leverage the spatial locality of dependencies in molecular structures for efficient computation. When evaluated on unseen folds, the model achieves significantly improved perplexities over the state-of-the-art parametric generative models. Our framework suggests the possibility of being able to efficiently design and engineer protein sequences with structurally-guided deep generative models, and underscores the central role of modeling sparse long-range dependencies in biological sequences.

ACKNOWLEDGMENTS

We thank members of the MIT MLPDS consortium for helpful feedback and discussions.

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

# 5 APPENDIX

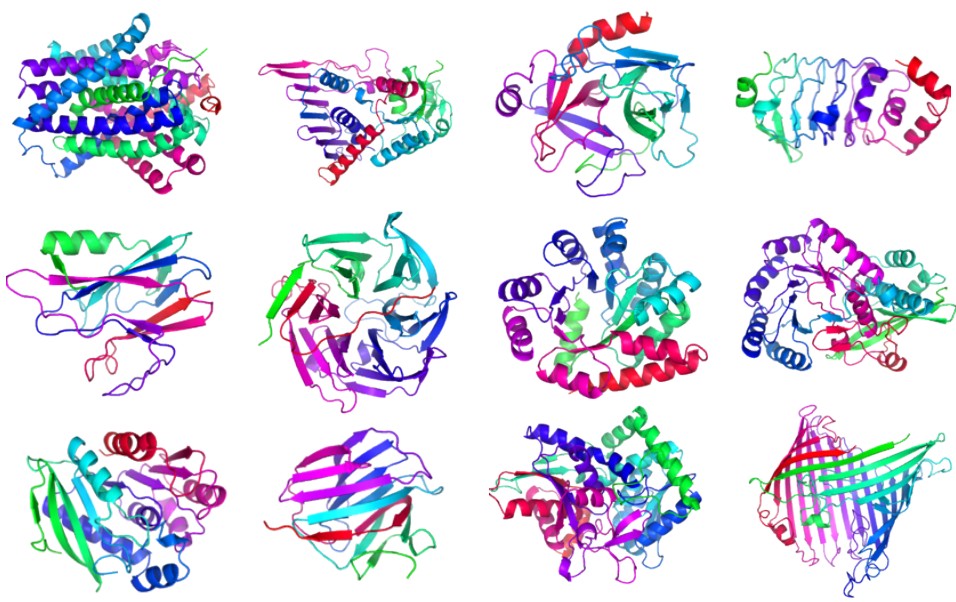

Figure 2: **Example structures from the dataset.**

