# OpenReview forum: "Generative Models for Graph-Based Protein Design"
_ICLR.cc/2019/Workshop/DeepGenStruct — DeepGenStruct 2019_

### Official Review · AnonReviewer2 · 2019-04-12
**Well written paper to an interesting & important application of generative models**

**Rating:** 4
**Confidence:** 2

**Review:**

This paper proposes to use an autoregressive Transformer model for the purpose of protein design. The model is well justified and overall the paper is written well. However there are couple of questions I have to authors:

1. Since protein is a graph that doesn't have a clear ordering, which ordering to you use in autoregressive decoder ?
2. Could you also compare the model with non-deep neural network baselines ?
3. How expensive would it be to evaluate the energy of generated proteins ?

---

### Official Review · AnonReviewer1 · 2019-04-15
**Interesting but experiments are lacking**

**Rating:** 2
**Confidence:** 2

**Review:**

This paper proposes a Transformer-based architecture for generating an amino acid sequence for a protein, given its 3D structure.  The authors define custom geometric features, and feed it to a model that has elements of a Transformer and graph convolutional neural network.

The main weakness is that the experiments section is limited:
- Direct comparison with graph convolutional neural networks is missing, despite this being a more standard way to do deep learning over graphs.
- There should be ablations of the different features explored, and perhaps comparisons to simpler featurization schemes that have been proposed in the past.
- There is a comparison with SPIN2, but there are some weird methodological issues, namely that pseudocounts were added post hoc to prevent infinite perplexity. There should be a way to fix numerical stability issues directly, without having to add these pseudocounts.
- RNN baselines seem to basically learn unigram frequencies, which either suggests they were not tuned properly or that they were too weak baselines, and some slightly better baselines should also be explored.
- The task of mapping structure to amino acid sequence was motivated by the goal of protein generation. However there is no actual evaluation of the generated sequences, only perplexity.

There were also some points of confusion:
- In 2.1, the authors mention that they can handle both "rigid backbone" and "flexible backbone" problems, but then exclusively discuss the rigid case. Since their featurization depends on having the 3D coordinates of all backbone amino acids, which seems to be a hallmark of the "rigid" setting, it is unclear how this extends to the "flexible" setting.
- It is unclear how the decoder works, especially because the j-th amino acid sequence is added to the edge features e_{ij}. This would seem to make it hard to use the standard masking trick to decode--do you have to recompute the entire set of features for each step of decoding?

---

### Decision · Program_Chairs · 2019-04-19
**Acceptance Decision**

**Decision:**

Accept

**Comment:**

The authors propose new model with geometric features for modeling 3d structure. There were some concerns with regard to clarity (e.g. how does decoding work?), which should be addressed for the camera-ready.